# PaDeLLM-NER: Parallel Decoding in Large Language Models for Named Entity Recognition

Jinghui Lu [1*], Ziwei Yang [1*], Yanjie Wang [1*], Xuejing Liu [2], Brian Mac Namee [3], Can Huang [1 ✉]

[1] ByteDance
[2] University of Chinese Academy of Sciences, China
[3] School of Computer Science, University College Dublin
{lujinghui, yangziwei.1221, wangyanjie.prince, can.huang}@bytedance.com
xuejing931210@gmail.com
brian.macnamee@ucd.ie

## Abstract

In this study, we aim to reduce generation latency for Named Entity Recognition (NER) with Large Language Models (LLMs). The main cause of high latency in LLMs is the sequential decoding process, which autoregressively generates all labels and mentions for NER, significantly increase the sequence length. To this end, we introduce **Pa**rallel **De**coding in **LLM** for **NER** (PaDeLLM-NER), a approach that integrates seamlessly into existing generative model frameworks without necessitating additional modules or architectural modifications. PaDeLLM-NER accelerates decoding by simultaneously generating all mentions at once, *i.e.*, a label-mention pair per sequence. This results in shorter sequences and faster inference. Experiments reveal that PaDeLLM-NER significantly increases inference speed that is 1.76 to 10.22 times faster than the autoregressive approach for both English and Chinese. Concurrently, it maintains the prediction quality as evidenced by the micro F-score that is on par with the state-of-the-art approaches under both zero-shot and supervised setting. All resources are available at `https://github.com/GeorgeLuImmortal/PaDeLLM_NER`.

## 1 Introduction

Named Entity Recognition (NER), a fundamental task in Natural Language Processing (NLP), aims to extract structured information from unstructured text data. This includes identifying and categorizing key elements such as Organization, Geopolitical Entity and so on (referred to as *"labels"*) in inputs, and pairing them with relevant text spans extracted from the text (termed *"mentions"*). Conventionally, NER tasks are carried out through an extractive paradigm that entails token-level classification and the subsequent extraction of identified tokens [1, 2].

Recent advancements in Large Language Models (LLMs) [8–13] have revolutionized numerous foundational tasks in NLP, including NER tasks [3–7, 14–17], through the adoption of a generative paradigm. This paradigm involves instruction-tuning a sequence-to-sequence (seq2seq) model. The model takes a sequence of unstructured text as input and produces a sequence of structured label-mention pairs as output. Generally, the output structured string should be formatted to meet two criteria: (1) it should have a clear and straightforward structure that facilitates post-processing for label and mention extraction, and (2) it needs to be generated fluidly and efficiently from the perspective of language models [18].

---

*  Equal contribution. ✉ Corresponding authors.

38th Conference on Neural Information Processing Systems (NeurIPS 2024).

| Variant | Input Unstructured Text | Output Structured Label-mention String |
|---------|------------------------|----------------------------------------|
| *Augmented Language* [3, 4] | Japan, co-hosts of the World Cup in 2002 and ranked 20th in the world by FIFA, are favourites to regain their title here. | [Japan \| LOC], co-hosts of the [World Cup \| MISC] in 2002 and ranked 20th in the world by [FIFA \| ORG], are favourites to regain their title here. |
| *Structured Annotation* [5–7] | Cuttitta announced his retirement after the 1995 World Cup , where he took issue with being dropped from the Italy side that faced England in the pool stages. | ((PER): (Cuttitta), (MISC): (1995 World Cup), (LOC): (Italy), (LOC): (England), (ORG): (NULL)) |

Table 1: Structured output string format used in the literature. The examples come from *CoNLL2003* dataset.

In Table 1, we list two typically used autoregressive output formats found in the literature : (1) accommodate original input text to contain label information, which is referred to as *"augmented language"* [3, 4]; (2) directly using a customized, easily-parsed structured format to output all labels and mentions, which is called *"structured annotation"* [5–7]. These formats present certain challenges. For example, *augmented language* necessitates duplicating all original input text, thereby increasing output length and resulting in inference inefficiency. While *structure annotation* avoids replicating the entire input, it produces all labels and mentions in an autoregressive manner. This implies that each subsequently generated pair depends on its preceding pairs, and when the number of label-mention pairs is large, it will lead to longer sequences. As demonstrated in Chen et al. [19], Ning et al. [20], high latency in LLMs mainly stems from lengthy sequence generation, we believe that by reducing the length of sequence, a more efficient inference scheme can be provided for NER tasks.

In light of this, we propose ***Pa**rallel **De**coding in **LLM** for **NER** (PaDeLLM-NER)*, a novel approach to accelerate the inference of NER tasks for LLMs. PaDeLLM-NER empowers the model with the capability to predict a single label-mention pair within a single sequence, subsequently aggregating all sequences to generate the final NER outcome. Specifically, in the training phase, we reconstruct the instruction tuning tasks, enabling LLMs to predict the count of mentions for a specific label and to identify the $n^{th}$ mention within the entire input for that label (Figure 1). In the inference phase, LLMs first predict the number of mentions for all labels, then predict all label-mention pairs in parallel (Figure 2). Finally, results from all sequences are aggregated and duplicate mentions across labels are eliminated based on prediction probability. This approach results in a more efficient inference method, producing shorter sequences and enabling parallel decoding label-mention pairs in batches.

Comprehensive experiments have been conducted, demonstrating that PaDeLLM-NER effectively reduces the number of tokens produced in each sequence, thereby decreasing inference latency. Additionally, it maintains or even enhances prediction quality in both flat and nested NER for English and Chinese languages, compared to existing methods in the literature under both zero-shot and supervised setting. To conclude, our contributions are as follows:

- We present PaDeLLM-NER, a novel approach tailored for NER using LLMs. This approach can predict all label-mention pairs in parallel, effectively reducing inference latency.

- Extensive experiments have been conducted, revealing that PaDeLLM-NER significantly improves inference efficiency. By completely decoupling the generation of label-mention pairs, the average sequence length is reduced to around 13% of that produced by conventional autoregressive methods. Correspondingly, the inference speed is 1.76 to 10.22 times faster than these previous approaches.

- Comprehensive experiments demonstrate that, in addition to its enhanced prediction speed, PaDeLLM-NER also maintains or surpasses the prediction quality of conventional autoregressive methods, on par with state-of-the-art (SOTA) performance on many NER datasets, including zero-shot as well as the supervised scenarios.

To the best of our knowledge, our technique stands as a pioneering approach in accelerating NER inference in LLMs by parallel decoding all label-mention pairs. This unique characteristic makes it complementary to other inference acceleration methods such as LLM.int8() [21] and speculative sampling [22, 23]. Thus, it can be efficiently integrated with these methods.

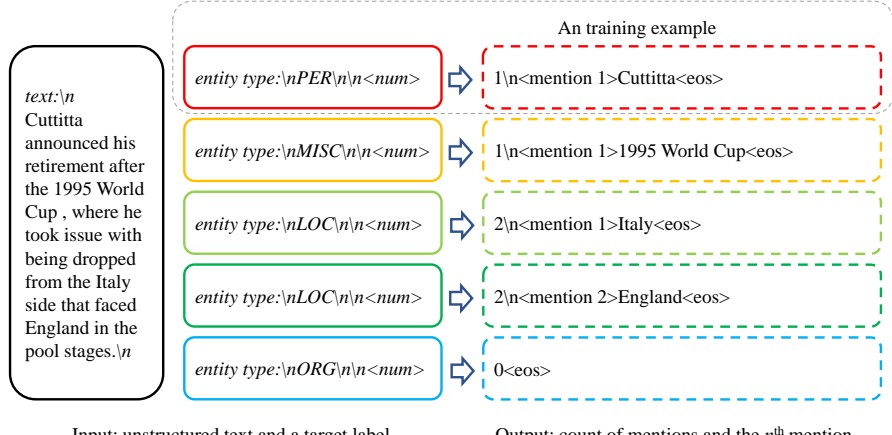

Input: unstructured text and a target label      Output: count of mentions and the $n^{th}$ mention

Figure 1: PaDeLLM-NER training paradigm: texts within frames of the same color represents one training example, where texts inside the solid-line frame are the input, and those inside the dashed-line frame are the output. *Italic* texts are prompt templates. The *"entity type"* signifies the label being predicted. The *"<num>"* indicates count of mentions for that label, and *"<mention n>"* refers to the $n^{th}$ mention of a label in the input.

## 2 Related Work

### 2.1 Generative Models for NER

Before the era of LLMs, most research approached NER as a sequence labeling task, where each token is assigned a pre-defined tag (*e.g.*, BIO scheme). In this line of work, usually pre-trained transformer-based language models [1, 2] is combined with a tailored prediction head to perform a token-level classification, followed by the extraction of identified tokens.

Encouraged by the success of unifying multiple NLP tasks into a single seq2seq paradigm [24, 25], especially with the evolution of LLMs [10, 13, 26, 27], the trend of applying seq2seq models to NER tasks is gaining momentum [28], with both inputs and outputs being represented as sequences of text [3–7]. Recently, the focus of work on NER using LLMs has shifted towards zero-shot [29, 30] or few-shot learning [4, 18, 31, 32], utilizing in-context learning [18, 32], self-consistency [29, 33] or learning programming [30, 34].

Unlike previous studies emphasizing few-shot performance with training-free prompt learning, our work focus on a fully supervised setting. More importantly, our primary objective is to speed up NER inference.

### 2.2 Inference Speedup in LLMs

Modern LLMs employ a sequential decoding strategy for token generation, which poses a significant challenge in terms of parallelization, especially as model size and sequence length increase [20]. There is plenty of work in the literature to address this challenge [35–38]. One line of work falls into training-free category such as introducing extra modules for speculative sampling [22, 23]. Another approaches explore modifying model architecture to accelerate inference, such as exiting at earlier layer [39, 40], or designing entirely different training and inference mechanisms [41–43]. Different from previous works, we focus on exploring the inference speedup in LLMs with a focus on the NER task without the change of model architecture or introducing extra modules.

## 3 Method

In this section, we delve into the details of PaDeLLM-NER. First, we focus on reframing the instruction tuning tasks as outlined in Section 3.1. Second, we explore the two-step inference process, detailed in Section 3.2. Finally, we discuss the aggregation of results and the technique for

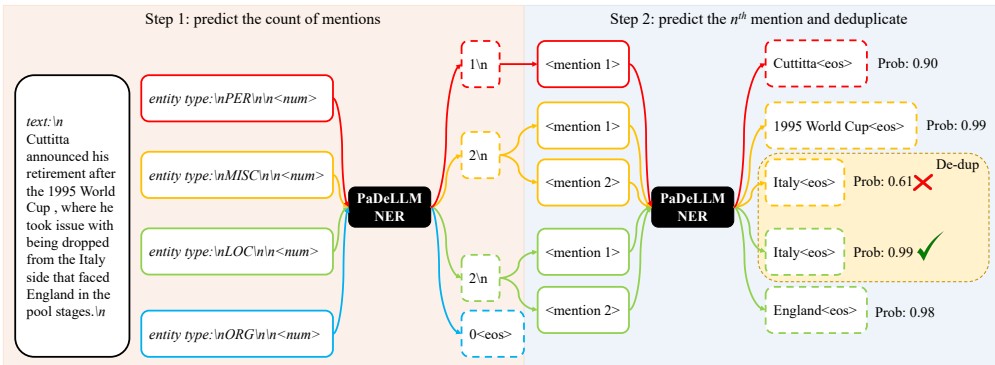

Figure 2: PaDeLLM-NER inference paradigm: texts enclosed in frames with identical colors indicate sequences of the same label. Specifically, the texts within solid-lined frames represent the added templates, while those within dashed-lined frames denote the prediction. In Step 1, the model predicts the number of mentions for all labels while in Step 2, it predicts the mentions. By aggregating mentions and labels from all sequences, the final NER results are obtained. Duplicate mentions appearing in different labels are resolved using prediction probabilities.

eliminating duplicate mentions across labels, which is elaborated in Section 3.3. An illustration of PaDeLLM-NER is shown in Figure 1 and Figure 2.

## 3.1 Reframing of Instruction Tuning

Illustration of the reframing is presented in Figure 1. As an example, we use a case from the *CoNLL2003* dataset including four labels: person (PER), miscellaneous (MISC), location (LOC), and organization (ORG). The specifics of the input text and the corresponding ground truth are provided in the second row of Table 1.

During reformulation, a single unstructured text containing all label-mention pairs is split into several sequences. Each new sequence's output includes the count of mentions for a specified label (denoted as *"entity type"*), followed by the $n^{th}$ mention of that label (denoted as *"<mention n>"*). Note that the count of mentions and their respective indices are represented using corresponding digit tokens from the LLM's vocabulary. Specifically, if there are no mentions, the model is trained to immediately predict the *"<eos>"* token, bypassing the need to predict mentions.

Therefore, in this example, one original training data is transformed into five new training data entries. These include two for predicting *"LOC"* (with 2 mentions), one for predicting *"MISC"* (with 1 mention), one for predicting *"PER"* (with 1 mention), and one for predicting *"ORG"* (with 0 mentions, directly predicting *"<eos>"*). Moreover, the number of mentions for each label and the text corresponding to each mention index can be easily obtained from the original ground truth, meaning that the number of new examples depends on the ground truth of that particular example.

With the newly reformulated training examples, we then apply the standard instruction tuning procedure. The model takes a sequence of text $t_1, t_2, \ldots, t_T$ consisting of input unstructured text and output structured label-mention pair. The optimization objective is cross-entropy loss $\mathcal{L}$ which can be defined as follows:

$$\mathcal{L} = -\frac{1}{T} \sum_{i=1}^{T} \log P\left(t_i \mid t_1, t_2, \ldots, t_{i-1}\right) \tag{1}$$

where $P\left(t_i \mid t_1, t_2, \ldots, t_{i-1}\right)$ represents the probability of $i^{th}$ token $t_i$ given the sequence of preceding tokens $t_1, t_2, \ldots, t_{i-1}$, as predicted by the model. Note that loss calculation begins from the number of mention tokens (*i.e.*, texts enclosed by dashed-line frames). Theoretically, loss from text spans such as *"<mention n>"* could be ignored during this calculation, since they simply prompt the mention's order, which does not necessarily need to be generated by the model. However, our

ablation studies show that ignoring these texts has negligible impact on model performance, a point further discussed in Appendix A. Therefore, we adhere to the standard instruction tuning procedure.

This reformulation allows the model to focus one label-mention pair at a time, shortening the generated length per sequence. More details are shown in Appendix C.

## 3.2 Inference of Label-Mention Pairs

Given a trained LLM, we propose a two-step inference approach: firstly, to predict the number of mentions for a specific label based on the prompt; and secondly, given the label and provided index to precisely identify the corresponding mention.

Figure 2 shows the overview of PaDeLLM-NER inference. In Step 1, the model predicts the total number of mentions for each label in the input, based on the label prompt. A separate token "\n" signals the completion of this count prediction. If no mentions of the given label exist, the model generates an "*<eos>*" token, skipping Step 2 for that label. In Step 2, following adding the predicted mention count to the input, mention indexes templates are appended. Formally, if the predicted number of mention is $m$, then "*<mention n>*", indicating the $n^{th}$ mention of the specified label, is appended for each $n$ within the set $\{1, 2, 3, \ldots, m\}$ and $n$ is an integer. Subsequently, the corresponding mention is generated by the model conditioned on preceding tokens. Note that the decoding of all label-mention pairs occurs in parallel, allowing for their simultaneous generation. Additionally, to justify the efficacy of the proposed two-step inference approach, we also implement a one-step parallel decoding method. In this approach, multiple mentions of the same label are predicted in a single sequence and compared to the two-step method in a preliminary experiment. Further details are provided in the Appendix A.

In practice, if there are sufficient GPU resources, the inference for the number of mentions for each label, as well as the subsequent inference for the mention text spans, can be allocating on separate GPUs. If GPU resources are limited, the inference can also be deployed on a single GPU using batch inference, facilitating parallel decoding. Using Figure 2 as an example, in Step 1, the batch size is four, as there are four labels in the dataset. In Step 2, the batch size is five, reflecting the five label-mention pairs determined in Step 1 (*i.e.*, 1 in *"PER"*, 2 in *"MISC"*, 2 in *"LOC"*). This parallel decoding strategy is effective in reducing inference latency, especially in scenarios where inputs are received in a streaming manner.

## 3.3 Removal of Duplicate Mentions

Unlike autoregressive decoding, where subsequent label-mention pairs can attend preceding ones, PaDeLLM-NER generates each label-mention pair independently. This inference strategy means that the model might generate mentions erroneously repeated in multiple labels. As exemplified in Figure 2, the model correctly predicts the first mention of *"LOC"* as *"Italy"*, but it also incorrectly predicts the second mention of *"MISC"* as *"Italy"*.

To address the issue of duplicate mentions, we suggest employing prediction probability to remove repeated mentions. Specifically, we calculate the prediction probability for each instance of the mention. This is done using the formula: $P = \prod_{i=b}^{e} P(t_i | t_1, t_2, \ldots, t_{i-1})$ where $b$ represents the starting token index of the mention text, and $e$ denotes the ending token index. Then, for a mention that appears in multiple labels, the mention instance with the highest probability will be preserved. As illustrated in Figure2, *"Italy"* is categorized as *"MISC"* with only a 0.61 probability, which is lower than that for *"LOC"*, resulting in its removal. In practice, the probability of each token can be calculated concurrently with token generation. Consequently, this method enables an efficient and accurate identification of duplicate mentions without incurring additional costs. The effectiveness of this de-duplication approach is further explored in Appendix A.

# 4 Experiments

In this section, we showcase the effectiveness of PaDeLLM-NER in terms of prediction quality and inference acceleration through experiments.

### 4.1 Setup

**Datasets**   The datasets used in our experiments include:

- **Zero-shot Datasets:** To align with the methodology proposed by [44], we train PaDeLLM using the Pile-NER dataset [45]. This dataset comprises around 240,000 entities categorized into 13,000 distinct types, derived from the Pile Corpus [46]. The passages in Pile-NER are enhanced through processing with ChatGPT, which facilitates the transparent generation of inherent entities. For assessing the model's zero-shot capabilities on previously unseen entity categories, following [30, 44, 45] we select two established benchmarks: CrossNER [47] and MIT [48].

- **Supervised Datasets:** we evaluate our method on supervised English and Chinese NER datasets. Following [30, 49, 50], English datasets include the general domain flat NER *CoNLL2003* [51], the nested NER *ACE2005* [52], and the biomedical nested NER *GE-NIA* [53]. Following [6, 54, 55], Chinese datasets include four commonly used general domain flat NER benchmarks *Resume* [56], *Weibo* [57], *MSRA* [58] and *Ontonotes 4.0* [59] and two vertical industrial domain flat NER datasets *YouKu* [60] and *Ecommerce* [61]. The statistics of all datasets are shown in Appendix B.

**Training setup**   We employ pre-trained version of Llama2-7b [11] and Baichuan2-7b [13] as base models for English and Chinese study respectively. Additional implementation details are in Appendix D.

**Inference setup**   For all generative models, we use greedy search with a beam size of 1, a maximum of 512 new tokens, and a temperature of 1.0. As described in Section 3.2, for PaDeLLM-NER, we adopt two inference settings: (1) each example is inferred on multiple GPUs to implement parallel decoding (*i.e.*, each sequence is assigned on one GPU), termed as **PaDeLLM$_{\text{Multi}}$**; and (2) each example is inferred on a single GPU, employing batch decoding for parallel decoding, termed as **PaDeLLM$_{\text{Batch}}$**. Note that for PaDeLLM$_{\text{Multi}}$, we sequentially predict each sequence of one example to simulate parallel decoding on multiple GPUs.

**Baselines**   The baseline used in our experiments include:

- **Inference Latency Baselines:** As the primary focus of this work is on reducing inference latency in NER tasks using LLMs, we compare our method, PaDeLLM-NER, with traditional autoregressive approaches. As mentioned in Section 1, the main points of comparison are autoregressive structured output formats used in [3, 4] and [5–7], referred to respectively as **AutoReg$_{\text{Aug}}$** and **AutoReg$_{\text{Struct}}$**, as these are the approaches very close to our system. We reimplemented all these methods for both English and Chinese datasets, utilizing the same pre-trained LLMs as in PaDeLLM-NER.

- **Zero-shot Baselines:** LLMs are known for their generalizability, therefore, following Ding et al. [44], we we also evaluate the zero-shot performance of PaDeLLM. Several most recent SOTA LLM-based approaches are selected as strong baselines as their great generalizability in zero-shot NER scenarios including **GoLLIE-7B** [30], **UniNER-7B** [45], **GLiNER-L** [62], **GNER-LLaMA-7B** [44].

- **Supervised Baselines:** We compare our approach with other recent SOTA supervised approaches, including **BINDER** [50], **Gollie** [30], and **DeepStruct** [49] for English benchmarks, as well as **W$^2$NER** [63], **NEZHA-BC** [54], and **SSCNN** [55] for Chinese benchmarks, to show PaDeLLM-NER's efficacy in prediction quality.

More details on the re-implementation and model size of each method are provided in Appendix D.

**Evaluation**   Our evaluation encompasses two dimensions: prediction quality and acceleration of NER inference. For assessing prediction quality, in line with Lu et al. [5], Wang et al. [7], we employ the micro F-score.

---

`https://huggingface.co/meta-llama/Llama-2-7b`
`https://huggingface.co/baichuan-inc/Baichuan2-7B-Base`

| | English Dataset | | | Chinese Dataset | | | | | | |
|---|---|---|---|---|---|---|---|---|---|---|
| **AutoReg** | **CoNLL03** | **ACE05** | **GENIA** | **Weibo** | **MSRA** | **Onto4** | **Resume** | **Youku** | **Ecom** | **Avg.** |
| AutoReg$_{Aug}$ | 992.70 | 944.90 | 1,515.35 | 1,276.32 | 812.78 | 1,009.68 | 982.39 | 579.99 | 845.42 | 995.50 |
| AutoReg$_{Struct}$ | 753.36 | 1,293.87 | 1,266.31 | 1,630.62 | 609.34 | 783.28 | 1,462.56 | 598.59 | 738.20 | 1,015.12 |
| **Ours** | | | | | | | | | | |
| PaDeLLM$_{Multi}$ | **229.74** | **255.53** | **316.90** | **159.57** | **143.47** | **171.67** | **238.27** | **203.63** | **293.40** | **223.57** |
| PaDeLLM$_{Batch}$ | 333.89 | 498.50 | 616.01 | 344.75 | 204.24 | 288.43 | 459.20 | 241.25 | 419.40 | 378.40 |

Table 2: Comparison of inference latency (in milliseconds) between PaDeLLM-NER and baseline methods. Underscored font is the second-best method, while a bold font is the best method, also applied to subsequent tables.

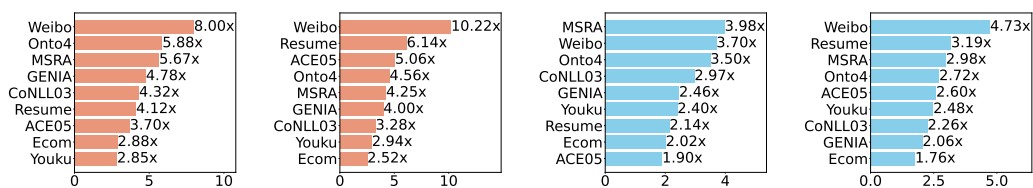

(a) AR$_{Aug}$ vs. PDLM$_{Multi}$  (b) AR$_{Struct}$ vs. PDLM$_{Multi}$  (c) AR$_{Aug}$ vs. PDLM$_{Batch}$  (d) AR$_{Struct}$ vs. PDLM$_{Batch}$

Figure 3: Speedup of PaDeLLM-NER compared to Autoregressive methods.

Following Ning et al. [20], we evaluate inference speed using latency (in milliseconds). We record the latency with the code: *start = time.time(); model.generate(); latency = time.time() - start*. In PaDeLLM-NER, we add the latency of mention counting and label-mention pair generation as the latency of each sequence. The final latency for the example is determined by the highest latency across sequences, as the user can only obtain the result of an example when the slowest sequence is generated. We conduct experiments three times and use the average result to alleviate the effect of randomness. We also report the average sequence length (tokenized) to clearly demonstrate the extent of sequence length reduction in Appendix E. Evaluations of all models were performed on the same NVIDIA A100 GPU.

## 4.2 Main Results

| Model | AI | Literature | Music | Politics | Science | Movie | Restaurant | Avg. |
|---|---|---|---|---|---|---|---|---|
| **SOTA** | | | | | | | | |
| GoLLIE-7B | 59.1 | 62.7 | 67.8 | 57.2 | _67.0_ | _63.0_ | 43.4 | 60.02 |
| UniNER-7B | 53.5 | 59.7 | 65.0 | 60.8 | 61.1 | 42.4 | 31.7 | 53.45 |
| GLiNER-L | 57.2 | 64.4 | _69.6_ | **72.6** | 62.6 | 57.2 | 42.9 | 60.92 |
| GNER-LLaMA-7B | **63.1** | **68.2** | **75.7** | _69.4_ | **69.9** | **68.6** | **47.5** | **66.05** |
| **Ours** | | | | | | | | |
| PaDeLLM-NER-7B | _60.7_ | _66.1_ | 67.6 | 68.1 | 64.4 | 61.3 | _43.6_ | _61.68_ |

Table 3: Comparison of prediction quality with recent SOTA models in zero-shot setting.

**Evaluation on inference latency** We investigate how PaDeLLM-NER reduces the end-to-end latency compared to baseline methods. Table 2 presents the average latency for each method across all datasets. First, it's clear that both PaDeLLM$_{Multi}$ and PaDeLLM$_{Batch}$ significantly reduce inference latency when compared to baseline methods, as highlighted by the substantial reduction in mean latency. For example, the mean latency reduction achieved between PaDeLLM$_{Multi}$ and AutoReg$_{Struct}$ stands at an impressive 791.55 ms, underscoring the significant improvement.

To more intuitively quantify the latency reduction of PaDeLLM-NER, we break down its speedup across different datasets in comparison to baseline methods in Figure 3. The speedup is computed by dividing the latency of baselines by the latency of PaDeLLM-NER. We can observe that PaDeLLM-

| SOTA | CoNLL03 | ACE05 | GENIA | Avg. |
|---|---|---|---|---|
| BINDER [50] | **93.33** | 89.50 | 80.50 | 87.77 |
| Gollie [30] | 93.10 | **89.60** | - | - |
| DeepStruct [49] | 93.00 | 86.90 | **80.80** | 86.90 |
| **AutoReg** | | | | |
| $AutoReg_{Aug}$ | 93.08 | 83.04 | 70.16 | 82.09 |
| $AutoReg_{Struct}$ | 91.87 | 82.99 | 77.90 | 84.25 |
| **Ours** | | | | |
| PaDeLLM-NER | 92.52 | 85.02 | 77.66 | 85.07 |

Table 4: Comparison of prediction quality with recent SOTA methods on English supervised datasets.

NER consistently show a speedup over baseline methods across all datasets. The highest speedup is observed in the *Weibo* dataset when comparing $AutoReg_{Struct}$ vs. $PaDeLLM_{Multi}$, with a speedup of 10.22x. When we narrow our focus to the comparison between $PaDeLLM_{Batch}$ and the baseline methods, considering these methods utilize a single GPU for inference, we can still observe substantial speedup ranging from 1.76x to 4.73x. The speedup factor varies across different datasets, suggesting that the efficiency gains of PaDeLLM-NER may be influenced by the characteristics of each dataset. Interestingly, we can observe that the $PaDeLLM_{Batch}$ is slower than $PaDeLLM_{Multi}$ (378.40 ms vs. 223.57 ms), more analysis about this is shown in Section 5.

Overall, the Table 2 and Figure 3 suggest that PaDeLLM-NER significantly reduces latency compared to autoregressive methods, though the extent of this reduction varies by dataset and the specific baseline method it's compared to.

**Evaluation on zero-shot prediction quality** Table 3 compares the prediction quality of different models across various domains like *AI*, *Literature*, *Music*, *Politics*, *Science*, *Movie*, and *Restaurant* in a zero-shot setting. Among all these methods, GoLLIE-7B scores range from 43.4 in *Restaurants* to 67.8 in *Music*, with an average of 60.02. UniNER-7B has lower scores, particularly in *Restaurants* (31.7), and averages 53.45. GLiNER-L shows a fairly balanced performance with a high of 72.6 in *Politics* and an average of 60.92. GNER-LLaMA-7B excels in *Music* with a 75.7 score and has the highest average of all at 66.05. Our model, PaDeLLM-NER, which consistently performs well across all domains. It has the second-best average score of 61.68, following GNER-LLaMA-7B. This highlights that while it is not the top performer, it offers robust and balanced prediction capabilities across a diverse set of topics in zero-shot setting. Note that the training of PaDeLLM-NER does not incorporate the additional task scheme prompt for describing unseen entities as used in GNER-LLaMA-7B [44], which may account for the observed differences in performance.

**Evaluation on supervised prediction quality** Table 4 and Table 5 present the micro F-scores of PaDeLLM-NER in comparison to other SOTA methods on supervised datasets. Notably, the micro F-scores for both $PaDeLLM_{Multi}$ and $PaDeLLM_{Batch}$ are identical. Initially, it is evident that encoder-based methods surpass LLM-based approaches, such as AutoReg and PaDeLLM-NER, within the supervised context. Nonetheless, the strength of LLM-based methods lies not in their performance under task-specific supervised settings, but rather in their superior zero-shot capabilities, which compensates for their relative shortcomings in supervised scenarios. Nevertheless, PaDeLLM-NER demonstrates SOTA performance on certain task-specific datasets, exemplified by its exceptional results on the *Youku* dataset.

Upon comparing PaDeLLM-NER with AutoReg, both of which are LLM-based methods, it becomes evident that PaDeLLM-NER outperforms AutoReg across both English and Chinese supervised datasets, as evidenced by its superior mean F-score. This outcome indicates that PaDeLLM-NER not only achieves lower inference latency but also maintains a higher level of prediction quality when contrasted with baseline methods.

In summary, the results presented in Table 2, 3, 4 and 5, demonstrate that our approach not only maintains superior prediction quality in both zero-shot and supervised environments but also significantly reduces inference latency.

| SOTA | Weibo | MSRA | Onto4 | Resume | Youku | Ecom | Avg. |
|---|---|---|---|---|---|---|---|
| NEZHA-BC [54] | - | - | - | - | - | **82.98** | - |
| SSCNN [55] | 71.81 | - | 82.99 | 96.40 | 86.10 | 81.80 | - |
| W$^2$NER [63] | **72.32** | **96.10** | **83.08** | **96.65** | - | - | - |
| **AutoReg** | | | | | | | |
| AutoReg$_{Aug}$ | 59.04 | 95.56[*] | 79.20 | 95.80 | 86.07 | 76.02 | 81.94 |
| AutoReg$_{Struct}$ | 56.07 | 90.92[*] | 80.97 | 95.74 | 86.85 | 81.57 | 82.02 |
| **Ours** | | | | | | | |
| PaDeLLM-NER | 67.36 | 95.03[*] | 80.81 | 94.98 | **87.91** | 81.85 | 84.66 |

Table 5: Comparison of prediction quality with recent SOTA methods on English supervised datasets. "*" indicates that results are not directly comparable.

# 5 Speedup Analysis

One concern noted is that batch inference does not speed up as much as inference distributed across multiple GPUs. This observation is consistent with our expectations and supported by Chen et al. [19] who found that batch inference in LLMs tends to be slower than single sequence inference under identical conditions, likely due to limitations in GPU memory bandwidth [64].

Transitioning from these performance considerations, it's noteworthy that PaDeLLM-NER is self-contained and can be seamlessly integrated with various generative architectures, including well-established decoder-only models [8–13] and recent innovations like RWKV [65], as well as multi-modal LLMs [66, 67] for tasks like Key Information Extraction tasks [68], all without needing architectural changes or additional data/modules. Also, it could be incorporated with off-the-shelf LLMs such as ChatGPT [27] and Claude-2 through prompt engineering without the need for further training, an aspect we plan to explore in future research.

# 6 Data Contamination Concerns

Since we are using LLMs as our foundational models, trained on extensive datasets from various online sources [11, 13], there is a chance that the models may have encountered parts of our evaluation sets during their pre-training phase, albeit unintentionally. This could potentially affect our experimental results. However, the primary focus of our experiments is the comparison of our proposed method with baseline methods. Given that these methods employ the same LLM as the base model, data contamination is unlikely to significantly impact the results.

# 7 Limitations

One clear disadvantage of PaDeLLM-NER is the multiplication of training examples from one to $m * n$, where $m$ is the label count and $n$ the mention count. Despite this, given that low latency is a major bottleneck in LLMs, trading longer training for lower latency is justifiable. Also, given the impressive generalization ability of LLMs, we believe that this method can be smoothly adapted to few-shot scenarios requiring less computation resources, which will be explored in future work.

Additionally, accurately counting the number of mentions remains a challenge for LLMs as discussed in Appendix F. This issue could be alleviated by implementing a specialized counting model dedicated to this task [69]. Another drawback is that reformulating label-mention pairs loses location information, which hinders tasks like downstream editing. We will address this in future work. Additionally, the de-duplication mechanism is overly aggressive, potentially removing mentions that can appear under different labels—a common issue in real-world applications (see Appendix A for more details).

Finally, there are several instances of re-computation within the pipeline that can be optimized. Specifically, input texts are encoded multiple times throughout the process. During batch decoding, certain

---

https://www.anthropic.com/news/claude-2

sequences may encounter the *"<eos>"* token earlier, but due to the nature of batch inference, these sequences continue to predict. We plan to improve this in the future by implementing enhancements like KV cache reuse and batch inference with an early quit mechanism, among other strategies.

## 8 Conclusion

In this study, we present PaDeLLM-NER, a parallel decoding framework for NER within LLMs. This approach enables batch parallel decoding of all label-mention pairs, significantly cutting down inference time by 1.76 to 10.22 times without sacrificing prediction accuracy.

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

# A  Ablation study

| Variant | CoNLL03 | ACE05 | GENIA | Mean |
|---------|---------|-------|-------|------|
| PaDeLLM-NER | **92.52** | 85.02 | **77.66** | **85.06** |
| + Loss ignoring | 92.01 | **85.18** | 73.47 | 83.55 |
| - De-duplication | 92.44 | 84.80 | 77.54 | 84.92 |
| + De-duplication$_{Reverse}$ | 92.38 | 84.44 | 77.38 | 84.73 |

Table 6: Ablations on ignoring loss and de-duplication.

In this section, we set out to investigate the effects of the different aspects of PaDeLLM-NER.

**Ignoring text spans in loss**  As discussed in Section 3.1, during training, it is permissible to overlook the loss of text span *"<mention n>"*, as the model does not need to generate this specific text, which is appended during inference. However, as shown in Table 6 illustrate, omitting these texts has minimal impact on prediction quality.

One possible explanation is that during training, the more significant challenge for LLMs lies in predicting the appropriate mention texts, rather than their format. As the model can readily learns to correctly position the format *"<mention n>"*, this aspect contributes minimally to the loss computation in training. In this case, computing the loss for all text is almost equivalent to "neglecting" the computation of loss for *"<mention n>"*.

**De-duplication**  To demonstrate the effectiveness of the de-duplication technique, we established two configurations as detailed in Table 6. The *-De-duplication* denotes the pipeline operating without the de-duplication technique; *+De-duplication$_{Reverse}$* indicates the pipeline that removes mentions with the highest probability, opposite to the original de-duplication technique.

Theoretically, PaDeLLM-NER should be the top-performing method, as its de-duplication eliminates noisy mentions, enhancing precision. Following closely is the *-De-duplication*, allows duplicate mentions to persist. *+De-duplication$_{Reverse}$* ranks lowest since it removes correct mentions and retains incorrect ones, lowering recall and precision simultaneously. As shown in Table 6, the results consistently align with our expectations, thereby verifying the effectiveness of the de-duplication process. Moreover, the difference among these variants is subtle, which can be attributed to the rare cases where duplicate mentions exist. This further highlights the robustness of proposed method.

We also report statistics in Table 7 and 8 showing that mentions under multiple labels are rare for both ground truth and PaDeLLM predictions. However, we recognize that the de-duplication mechanism can be overly aggressive, potentially removing mentions that appear under multiple labels—a common scenario in real-world applications. In such cases, opting not to use the de-duplication mechanism may be preferable.

| Dataset | Count | Ratio |
|---------|-------|-------|
| ACE05 | 1 | 0.00034 |
| ConLL03 | 1 | 0.00017 |
| GENIA | 0 | 0 |
| Ecom | 0 | 0 |
| MSRA | 1 | 0.00013 |
| Weibo | 0 | 0 |
| Youku | 2 | 0.0012 |
| Resume | 0 | 0 |

Table 7: Mentions appear under multiple labels in ground truth.

**Preliminary experiments for justifying the importance of two-step prediction**  We conducted preliminary experiment using one-step prediction, where all mentions of the same label are predicted in a single sequence, which is referred to as *OneStep* in this paper. An example of OneStep parallel decoding is shown in Table 9. Note that the order of mentions is preserved as in the ground truth,

| Dataset | Count | Ratio |
|---------|-------|-------|
| ACE05 | 22 | 0.0074 |
| ConLL03 | 10 | 0.0017 |
| GENIA | 18 | 0.0034 |
| Ecom | 2 | 0.0012 |
| MSRA | 5 | 0.00089 |
| Weibo | 0 | 0 |
| Youku | 3 | 0.0019 |
| Resume | 0 | 0 |

Table 8: Mentions appear under multiple labels in PaDeLLM prediction.

following the data from the corresponding dataset. The overall latency for each example is determined by the latency of the slowest sequence. The preliminary experiment is conducted on three English dataset, i.e., CoNLL03, ACE05 and GENIA.

| Entity | Text | NER Result |
|--------|------|------------|
| ORG | <entity>ORG<text>2004-12-20T15:37:00 Microscopic microcap Everlast , mainly a maker of boxing equipment , has soared over the last several days thanks to a licensing deal with Jacques Moret allowing Moret to buy out their women 's apparel license for $ 30 million , on top of a $ 12.5 million payment now . | ["Microscopic microcap Everlast", "a maker of boxing equipment", "their"] |
| PER | <entity>PER<text>2004-12-20T15:37:00 ... million payment now. | ["Jacques Moret", "Moret", "their", "their women"] |
| GPE | <entity>GPE<text>2004-12-20T15:37:00 ... million payment now. | [] |
| LOC | <entity>LOC<text>2004-12-20T15:37:00 ... million payment now. | [] |

Table 9: Illustration of one-step parallel decoding NER approach.

| Method | ACE05 | CoNLL03 | GENIA | Mean |
|--------|-------|---------|-------|------|
| PaDeLLM$_{Multi}$ | **255.53** | **229.74** | **316.90** | **267.39** |
| OneStep$_{Multi}$ | 386.93 | 272.22 | 513.63 | 390.93 |
| AutoReg$_{Aug}$ | 944.90 | 992.70 | 1,515.35 | 1150.98 |
| AutoReg$_{Struct}$ | 1,293.87 | 753.36 | 1,266.31 | 1104.51 |

Table 10: Comparison of inference latency.

| Method | ACE05 | CoNLL03 | GENIA | Mean |
|--------|-------|---------|-------|------|
| PaDeLLM$_{Multi}$ | **85.02** | 92.52 | 77.66 | **85.06** |
| OneStep$_{Multi}$ | 80.98 | 91.36 | 76.27 | 82.87 |
| AutoReg$_{Aug}$ | 83.04 | **93.08** | 70.16 | 82.09 |
| AutoReg$_{Struct}$ | 82.99 | 91.87 | **77.90** | 84.25 |

Table 11: Comparison of prediction quality.

The results are reported in Table 10 and Table 11. As expected, the inference speed of one-step approach falls between that of the two-step prediction (i.e., PaDeLLM) and the purely autoregressive model. However, the prediction quality is lower compared to the two-step prediction. In other words, PaDeLLM outperforms the one-step approach in both inference speed and prediction quality, which again verifies the efficacy of PaDeLLM.

**Preliminary experiments for zero-shot autoregressive baseline**   We did not report zero-shot results for AutoReg_aug and AutoReg_struct as they are unsuitable for this setting. Preliminary experiments show higher latency and lower F-scores compared to PaDeLLM (see Table 12 for details).

| | AI | Literature | Music | Politics | Science | Avg |
|---|---|---|---|---|---|---|
| **Latency (ms)** | | | | | | |
| PaDeLLM | 398.37 | 357.45 | 352.85 | 366.76 | 375.02 | 370.09 |
| Auto_Aug | 1529.95 | 2096.08 | 2545.20 | 2364.87 | 2334.05 | 2174.03 |
| **F-score** | | | | | | |
| PaDeLLM | 60.7 | 66.1 | 67.6 | 68.1 | 64.4 | 65.38 |
| AutoReg_Aug | 0.19 | 0.15 | 0.94 | 0.13 | 0.21 | 0.324 |

Table 12: Comparison of Latency and F-score between PaDeLLM and AutoReg_Aug under zero-shot scenarios.

# B   Dataset Statistics

| Variant | CoNLL03 | ACE05 | GENIA | Mean |
|---|---|---|---|---|
| PaDeLLM-NER | 92.52 | 85.02 | 77.66 | 85.06 |
| + Model scale up to 13B | **93.02** | 84.37 | **78.84** | **85.45** |

Table 13: Ablations on model scaling up.

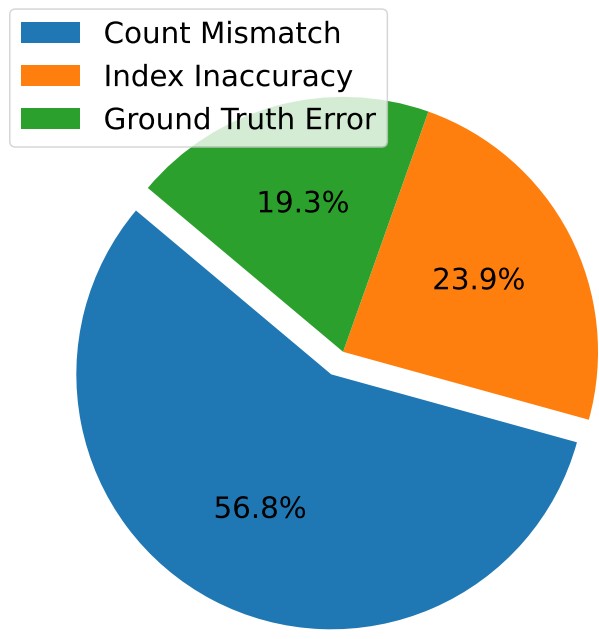

Figure 4: Percentage of different error types.

We evaluate our framework on 3 English and 6 Chinese flat/nested NER datasets. In Table 15, we present the detailed statistics. Note that while the statistics of the development set are reported, our training process does not involve the development set.

For the *MSRA* dataset, we excluded four outlier instances from the test set due to their excessively high number of names, significantly deviating from typical examples. These outliers not only posed challenges for model inference but also risked distorting the evaluation metrics, potentially leading to an inaccurate assessment of the model's performance on representative data.

| AutoReg | English Dataset | | | Chinese Dataset | | | | | | |
|---|---|---|---|---|---|---|---|---|---|---|
| | CoNLL03 | ACE05 | GENIA | Weibo | MSRA | Onto4 | Resume | Youku | Ecom | Mean |
| AutoReg$_{Aug}$ | 33.85 | _37.10_ | 60.50 | _45.02_ | 27.42 | 35.90 | _30.39_ | _18.21_ | 31.50 | _35.54_ |
| AutoReg$_{Struct}$ | _28.36_ | 49.95 | _49.03_ | 62.45 | _18.97_ | _25.53_ | 53.02 | 18.56 | _22.51_ | 36.48 |
| **Ours** | | | | | | | | | | |
| PaDeLLM-NER | **6.54** | **8.29** | **10.05** | **2.19** | **2.23** | **2.68** | **4.87** | **3.66** | **3.27** | **4.86** |

Table 14: Comparison of the number of generated tokens per sequence by PaDeLLM-NER with baseline methods.

Also, we perform label mapping to convert ground truth from special tokens to Chinese words following [6]. Further details are provided in Table 16.

| Dataset | Sentence | | | | Mention | | | |
|---|---|---|---|---|---|---|---|---|
| | #All | #Train | #Dev | #Test | #All | #Train | #Dev | #Test |
| CoNLL2003 | 20,744 | 14,041 | 3,250 | 3,453 | 35,089 | 23,499 | 5,942 | 5,648 |
| ACE2005 | 9,210 | 7,194 | 969 | 1,047 | 30,634 | 24,441 | 3,200 | 2,993 |
| GENIA | 18,546 | 15,023 | 1,669 | 1,854 | 56,015 | 46,142 | 4,367 | 5,506 |
| Weibo | 1,890 | 1,350 | 270 | 270 | 2,701 | 1,894 | 389 | 418 |
| MSRA$^{*}$ | 50,725 | 44,364 | - | 4,361 | 80,214 | 74,703 | - | 5,511 |
| OntoNotes 4.0 | 24,371 | 15,724 | 4,301 | 4,346 | 28,006 | 13,372 | 6,950 | 7,684 |
| Resume | 4,759 | 3,819 | 463 | 477 | 16,565 | 13,438 | 1,497 | 1,630 |
| Youku | 10,002 | 8,001 | 1,000 | 1,001 | 15,905 | 12,754 | 1,581 | 1,570 |
| Ecommerce | 4,987 | 3,989 | 500 | 498 | 15,216 | 12,109 | 1,540 | 1,567 |

Table 15: Dataset Statistics. "#" denotes the amount. For *MSRA*, we remove four outlier examples in test set.

| Dataset | #Entity | Entity |
|---|---|---|
| Weibo | 8 | {"PER.NAM(Specific Name)":"名称特指", "PER.NOM(Generic Name)":"名称代称", "GPE.NAM(Specific Geo-Political Entity)":"行政区特指", "GPE.NOM(Generic Geo-Political Entity)":"行政区代称", "LOC.NAM(Specific Location)":"地点特指", "LOC.NOM(Generic Location)":"地点代称", "ORG.NAM(Specific Organization)":"组织特指", "ORG.NOM(Generic Organization)":"组织代称" } |
| MSRA | 3 | {"LOC":"地点', "PER":"名称", "ORG":"组织"} |
| OntoNotes 4.0 | 4 | {"GPE":"地缘", "LOC":"地点", "PER":"名称", "ORG":"组织"} |
| Resume | 8 | {"NAME":"名称", "CONT(Nationality)":"国籍", "RACE":"民族", "TITLE":"职位", "EDU":"学历", "ORG":"公司", "PRO(Profession)":"专业", "LOC(Place of Birth)":"籍贯"} |
| Youku | 3 | {"TELEVISION":"电视剧", "PER(Celebrity)":"明星", "MISC":"其他"} |
| Ecommerce | 2 | {"HP(brand)":"品牌", "HC(commodity)":"商品"} |

Table 16: Entity tag of each dataset and the conversion from tag used in dataset to corresponding Chinese natural language. For some tags that are hard to understand, we provide their meaning in brackets. "#" denotes the amount of entity types.

## C Reformulation Examples

Two compete reformulated examples are presented in Table 17 for English and Chinese, respectively.

| Language | Input | Output |
|----------|-------|--------|
| *English* | text:
But Fischler agreed to review his proposal after the EU 's standing veterinary committee , mational animal health officials , questioned if such action was justified as there was only a slight risk to human health .
entity type:
PER
<num> | 1
<mention 1>Fischler |
| *Chinese* | 文本(text):
公报最后说，墨西哥政府认为，贩毒以及洗钱等与毒品有关的活动是威胁到国家主权和安全的一个全球性问题。(The communique concluded by stating that the Mexican government considers drug trafficking and related activities such as money laundering to be a global issue that threatens national sovereignty and security.)
指定NER标签(entity type):
地点(LOC)
<数量>(<num>) | 1
<第1文段>(<mention 1>) 墨西哥(Mexican) |

Table 17: Reformulated examples for English and Chinese dataset, respectively. We provide translations to facilitate understanding. The examples come from *CoNLL2003* and *MSRA* dataset.

# D Implementation Details

We train our model on all datasets for $4$ epochs, using a batch size of $128$ and a learning rate of $1e-5$, with the AdamW optimizer [70] and a cosine scheduler [71]. The maximum input and output sequence lengths are set to 2048 and 512, respectively. Training is conducted on 8 NVIDIA A100 GPUs. This configuration is applied across all PaDeLLM-NER models, as well as three baseline models: AutoReg$_{Aug}$, AutoReg$_{Struct}$ as well as Onestep baseline reported in preliminary experiment. We also report the model size of each NER method in Table 18

| English Method | Base Language Model | Chinese Method | Base Language Model |
|----------------|---------------------|----------------|---------------------|
| BINDER | BERT-base 110M | NEZHA-BC | NEZHA-base 110M |
| Gollie | Code-llama 34B | SSCNN | not report |
| DeepStruct | GLM10B | W2NER | Transformer-based 110M |
| AutoRegAug | LLaMA-2-7B | AutoRegAug | Baichuan2-7B |
| AutoRegStruct | LLaMA-2-7B | AutoRegStruct | Baichuan2-7B |
| PaDeLLM-NER | LLaMA-2-7B | PaDeLLM-NER | Baichuan2-7B |

Table 18: Model size of each NER method.

# E Sequence Length Reduction

Results of average sequence length produced by different approaches are presented in Table 14. Most notably, PaDeLLM-NER generates much shorter sequences than the other models across all datasets. The lengths range from 6.54 on *CoNLL20023* to 10.05 on *GENIA* for English datasets, and from 2.19 on *Weibo* to 4.87 on *Resume* for Chinese datasets. The mean length for PaDeLLM-NER is 4.86, which is significantly lower than the means of the other approaches: 35.54 for AutoReg$_{Aug}$ and 36.48 for AutoReg$_{Struct}$.

In summary, the result shows that PaDeLLM-NER produces much shorter generated sequences compared to the other methods, which is around 13.19% to 13.67% of the original length, respectively, indicating higher efficiency in its inference.

# F  Error analysis

**PaDeLLM-NER error analysis**    For our error analysis, we utilize the *ACE2005* dataset. We sample and manually examine 50 erroneous examples for analysis. We seek to identify the root causes of errors, which we have categorized into three types: (1) incorrect mention count, referred to as *Count Mismatch*; (2) inaccuracies in the mention corresponding to a specific index, termed *Index Inaccuracy*; and (3) errors in the ground truth data, known as *Ground Truth Errors*.

The distribution of each error type is illustrated in Figure 4. It is important to note that a significant portion of the errors stem from inaccuracies in mention counts (*i.e.*, Count Mismatch, about 56.8%), underscoring the necessity for enhancements in the model's counting capabilities. Accurate mention counts are pivotal for the quality of predictions. Overestimating the mention count often leads the model to either repeat the last entity or, more problematically, fabricate an entity, thereby escalating the rate of false positives. Conversely, underestimating the mention count results in the model's inability to identify some entities, thus increasing the incidence of false negatives. Following closely is the Index Inaccuracy error, indicating that the model sometimes struggles to accurately pinpoint the correct mention for a given index, further emphasizing areas for improvement.

Interestingly, our analysis reveal that a significant portion of the model's predictions, specifically 19.3%, are actually correct, challenging the accuracy of the ground truth data. This observation suggests the presence of inaccuracies within the ground truth, contributing to an elevated rate of false positives. Prior research, as noted in studies by Min et al. [72], Wang et al. [73], Zhou et al. [74], has demonstrated that LLMs predominantly acquire their knowledge during the pre-training phase. These models develop certain "core beliefs" that tend to align more closely with human judgment. In this context, it appears that the models possess an inherent capability to rectify errors in the ground truth data, demonstrating their potential to improve data accuracy beyond initial human annotation.

# G  Model Scaling Up

As we increase the model size to 13B, Table 13 presents a mix of results. In datasets like *CoNLL2003* and *GENIA*, the model shows a significant improvement in predictions. In contrast, the results on *ACE2005* are slightly worse. Note that the improvement in *GENIA* is substantial, at approximately 1.18%. Based on these findings, it seems reasonable to suggest that continuously scaling up the model size has the potential to maintain the performance that is at least on par, or even superior, especially in specific industrial domains. However, this hypothesis warrants further investigation, involving more families of models [8–13] and a broader range of datasets. We leave this exploration for future work.

