# OpenReview forum: "PaDeLLM-NER: Parallel Decoding in Large Language Models for Named Entity Recognition"
_NeurIPS.cc/2024/Conference — NeurIPS 2024 poster_

### Official Review · Reviewer_1fG5 · 2024-07-12

**Soundness:** 3
**Presentation:** 2
**Contribution:** 2
**Rating:** 6
**Confidence:** 4

**Summary:**

This paper introduces a novel approach to reduce generation latency in Named Entity Recognition (NER) using Large Language Models (LLMs). The primary issue addressed is the high latency caused by the sequential decoding process in LLMs, which significantly lengthens the sequence by autoregressively generating all labels and mentions for NER. To tackle this, the authors propose Parallel Decoding in LLM for NER (PaDeLLM-NER), which integrates into existing generative model frameworks without requiring additional modules or architectural changes. PaDeLLM-NER enables simultaneous decoding of all mentions, effectively reducing generation latency. Experimental results show that PaDeLLM-NER can improve the inference speed than the traditional autoregressive approach.

**Strengths:**

- The parallel decoding strategy is well-designed and experimental results prove the effectiveness.
- The authors provide comprehensive experiments with different setting and with furthre analysis.
- The paper is easy to follow.

**Weaknesses:**

- The proposed method cannot improve the inference speed in scenarios where only one type of entity is predicted.
- Since the method focuses on the inference efficiency of LLMs-based NER, it is better to report both inference speed and performance compared to zero-shot (Table 3) and supervised (Tables 4 and 5) methods. Notably, Table 3 only reports performance without considering the efficiency of different LLMs. Furthermore, why not report the performance of AutoReg_aug and AutoReg_struct in Table 3?
- For better understanding of the training resource usage when compared with other methods, it is better to report the base language models used (SOTA methods) in Tables 4 and 5.
- The writing of this paper could be further improved. For example, Line 219, “As per Ning et al...” appears to be a typo; the meanings of the underline (second performance) and bold (best performance) are not provided; and there is no explanation for why “*” indicates that results are not directly comparable in the Table 5 caption.
- Comparing with fixed few-shot in-context learning of LLMs may also be worth considering, as caching the fixed prompt could improve the inference speed of LLMs.

**Questions:**

Please see the weakness.

**Limitations:**

The authors provide one limitation section.

---

> ### Author Rebuttal · Authors · 2024-08-05
>
> We thank the reviewer for the thoughtful feedback and appreciate the recognition of our paper’s novelty and writing. We are grateful for the opportunity to address the concerns raised.
>
> **W1-speedup is weak when only one entity type**: Even if the entity type is one, if there are many mentions in that entity type, PaDeLLM can still improve inference speed through Step 2, because we predict all label-mention pair in parallel. As compared in the table,  the string length of PaDeLLM still shorter than AR method (<mention x> is the templated added not generated by the model).
>
>
> |                   | Prediction                                                                                                                                                      |
> |-------------------|-----------------------------------------------------------------------------------------------------------------------------------------------------------------|
> | Autoregressive    | str1: "LOC: England, India, China, the US"                                                                                                                      |
> | PaDeLLM           | str1: "<mention 1>England"
> |         | str2: "<mention 2>India"
> |            |  str3: "<mention 3>China"
> |          |  str4: "<mention 4> the US"                                         |
>
>
> **W2-reporting other method latency**:
>  - Our primary comparison for inference efficiency is between two typical AR methods, AutoReg_aug and AutoReg_struct. Other methods, such as BINDER, are encoder-only models and are not comparable. Methods like GoLLIE and DeepStruct use larger LLMs and different prompt engineering, making direct inference speed comparisons unfair.
> - We did not report the results for AutoReg_aug and AutoReg_struct in Table 3 because these methods are not immediately suitable for the zero-shot setting. Our preliminary experiments show that they have significantly higher latency compared to PaDeLLM, and their F-score is quite limited. We will include these experimental results in the updated version.
>
> | Latency  | AI     | Literature | Music  | Politics | Science | Avg    |
> |----------|--------|------------|--------|----------|---------|--------|
> | PaDeLLM  | 398.37 | 357.45     | 352.85 | 366.76   | 375.02  | 370.09 |
> | Auto_Aug | 1529.95| 2096.08    | 2545.20| 2364.87  | 2334.05 | 2174.03|
>
> | F-score  | AI   | Literature | Music | Politics | Science | Avg   |
> |----------|------|------------|-------|----------|---------|-------|
> | PaDeLLM  | 60.7 | 66.1       | 67.6  | 68.1     | 64.4    | 65.38 |
> | Auto_Aug | 0.19 | 0.15       | 0.94  | 0.13     | 0.21    | 0.324 |
>
>
> **W3-report base model**: We have added a column specifying the base language model used by each method and will update these tables in the next version. Except for GoLLIE and DeepStruct, all other methods are encoder-only, which results in a smaller number of parameters. GoLLIE and DeepStruct use backbones of 34B and 10B parameters, respectively, which are larger than PaDeLLM based on LLaMA-2 with 7B parameters.
>
> | Method        | Base Language Model |
> |---------------|---------------------|
> | BINDER        | BERT-base 110M      |
> | Gollie        | Code-llama 34B      |
> | DeepStruct    | GLM10B              |
> | AutoRegAug    | LLaMA-2-7B          |
> | AutoRegStruct | LLaMA-2-7B          |
> | PaDeLLM-NER   | LLaMA-2-7B          |
>
> | Method        | Base Language Model     |
> |---------------|-------------------------|
> | NEZHA-BC      | NEZHA-base 110M         |
> | SSCNN         | not report              |
> | W2NER         | Transformer-based 110M  |
> | AutoRegAug    | Baichuan2-7B            |
> | AutoRegStruct | Baichuan2-7B            |
> | PaDeLLM-NER   | Baichuan2-7B            |
>
>
> **W4-writing improvement**
>
>   - Thank you for pointing out the typos. We will correct them in the updated version.
>   - The meanings of underline and bold formatting will be clarified in the updated version.
>   - The reason for the asterisk (*) is explained in  the caption of Table 9. We will provide a more explicit explanation in the updated version.
>
> **W5-considering ICL**: This is a good suggestion. We have acknowledged it in lines 287-289, and we will rephrase the wording to make it clearer. We'll also list caching mechanism as a potential area for future exploration (line 310).

---

> ### Comment · Reviewer_1fG5 · 2024-08-09
> **To Authors**
>
> Thanks for your responses.
>
> > W1-speedup is weak when only one entity type...
>
> This may need to be acknowledged in your paper. Since only one mention also existing in many scenarios.
>
> > W3-report base model...
>
> Thanks your adding this column. This is very important for readers to understand the cost of reaching the coressponding performance.
>
>
> Though the decoding techniques could improve the efficiency. I still believe that using caching techniques and long-context ICL, i.e., fixed 1000-shots, may reach better performance but good efficiency. But I acknowledge that these techniques could be used in the decoding methods propoed in this paper.
>
> Since most of my concerns have been addressed. I increase my soundness score to 3 and over accessment score to 6.

---

> > ### Author Response · Authors · 2024-08-09
> > **Thanks**
> >
> > Thank you for the thoughtful review. We will address the speedup weakness and other necessary modifications in the revised version.

---

### Official Review · Reviewer_14Sq · 2024-07-13

**Soundness:** 3
**Presentation:** 3
**Contribution:** 3
**Rating:** 9
**Confidence:** 4

**Summary:**

They create an NER system where an LLM first outputs the number of mentions there are of a given type (for all possible types). Then all mentions can be generated in parallel.

This results in faster inference times as each generation is short, and they can be done in parallel.

**Strengths:**

They compare to several different baseline on multiple NER datasets in multiple different settings.

Their method is much faster than others.

**Weaknesses:**

Their reformulation of NER as predicting (label, mention) pairs removes a critical component of classical NER, the actual alignment of the mention to the tokens. Polysemous words are often mentions in some context and not in others and it if often important to know which one was the actual mention, especially if it is used for things like editing downstream.

The deduplication strategy is very aggressive and removes the possibility that some surface text is a label for multiple types in a single sentence. For example, "It is England vs. Italy on this sunny day in England", England is both a place (LOC) and a sports team (ORG) this would get filtered  by their setup.

The prose's definition of " prediction quality [...] that is on par" is rather loose, with their model being 6 points behind on average for zero-shot (table 3) and behind by a point of two on most supervised datasets.

**Questions:**

How do you expect this to scale to NER datasets like Ontonotes where a there are 20+ different mention category? Similarly what about long documents that could have 10-30+ mentions of a given type?

Did you see inconsistencies in the model outputs? For example a model that output that there is `1` person but then generated <mention 2> ${person name}?

**Limitations:**

yes

---

> ### Author Rebuttal · Authors · 2024-08-05
>
> We thank the reviewer for the thoughtful feedback and appreciate the recognition of our paper’s contributions. We are grateful for the opportunity to address the concerns raised.
>
> **W1-loss aligment of mention to the tokens**: We acknowledge that PaDeLLM loses token position information, and we recognize that incorporating token location (like start and end index) might enhance performance. We will consider this in future work, as suggested in [1].
>
> [1] A Unified Generative Framework for Various NER Subtasks (Yan et al., ACL-IJCNLP 2021)
>
> **W2-dedpulication is aggressive**: Yes, we acknowledge the de-duplicate strategy is aggressive. In our dataset, instances where two mentions appear under multiple labels are rare, as evidenced by our statistics. This is why we propose using the de-duplication mechanism. However, in real-world applications where a mention might be allowed to appear under multiple labels, we can choose not to use the de-duplication mechanism. This decision is a trade-off.
>
> | Dataset | Count | Percentage |
> |---------|-------|------------|
> | ACE05   | 1     | 0.00034    |
> | ConLL03 | 1     | 0.00017    |
> | Genia   | 0     | 0          |
> | ecom    | 0     | 0          |
> | msra    | 1     | 0.00013    |
> | weibo   | 0     | 0          |
> | youku   | 2     | 0.0012     |
> | resume  | 0     | 0          |
>
> **W3-improper wording**: We will tone down the wording in the updated version.
>
> **Q1-large number of entity types or mentions**:
>
> - Theoretically, this method is effective regardless of the number of mention categories because each label-mention pair is handled by a different string. If you have more than 20 different mention categories, you can process them in parallel using over 20 different sequences to predict the number of mentions. Once that is complete, proceed to Step 2 to predict the label-mention pairs.
> - In our in-house NER tasks, where there are often numerous mentions of a given type, we found that the method can manage tens of mentions without significantly losing accuracy. However, in practice, we typically limit the input to 512 tokens to prevent potential issues.
>
> **Q2-generating inconsistency**: During inference,  <mention x> is added to the prompt, which is not generated by the model. For instance, if the model outputs that there is `1` person at Step 1, we include <mention 1> in the prompt. If the model indicates there are  `2` person at Step1, we duplicate the input string and add  <mention 1> and <mention 2> to two strings, respectively. This approach effectively avoids inconsistencies. See Figure 2 caption the dashed box vs the solid box, and line 139-144.

---

### Official Review · Reviewer_XzjF · 2024-07-14

**Soundness:** 3
**Presentation:** 3
**Contribution:** 3
**Rating:** 6
**Confidence:** 3

**Summary:**

This paper presents PaDeLLM-NER, a novel approach for accelerating Named Entity Recognition (NER) inference in Large Language Models (LLMs) through parallel decoding. A reformulation of the NER task that enables parallel generation of label-mention pairs, significantly reducing inference latency. A two-step inference process involving mention count prediction and parallel mention generation.
Extensive experiments demonstrated significant speedups (1.76x to 10.22x faster) compared to autoregressive approaches while maintaining or improving prediction quality across multiple datasets and two languages.

**Strengths:**

The parallel decoding strategy for NER is innovative and addresses a significant bottleneck in LLM inference speed, which is important in some speed-sensitive applications. The authors conduct extensive experiments across multiple datasets, languages, and settings (zero-shot and supervised), proving the method's effectiveness. The reported speedups are substantial and could have a meaningful, practical impact on NER applications. The method is compatible with existing LLM architectures and can be integrated with other acceleration techniques. The methodology is well-explained with helpful diagrams and examples.

**Weaknesses:**

Some details and corner cases are not well explained. For example, I didn't see the token location information in Figure 2. If the input has multiple and same mentions (e.g., "Donald Trump owns the Trump Organization" ), how does this framework distinguish with the same mentions? (e.g. Trump in the above example)

In addition, it is not clear how the de-duplicate model processes the partially duplicated mentions. For example, in the above case,  the "Trump organization" was recognized as ORG, and what if the person module predicted the "Trump" in the "Trump organization" as a person? Will the de-duplicate model filter this case?

**Questions:**

No

---

> ### Author Rebuttal · Authors · 2024-08-05
>
> We thank the reviewer for the thoughtful feedback and appreciate the recognition of our paper’s contributions, writing and novelty. We are grateful for the opportunity to address the concerns raised.
>
> **W1-token location information**:
>   - NER traditionally relies on sequence labeling, where each token is assigned a class based on its position. This requires token location information. However, in the seq2seq approach such as PaDeLLM, Autoregressive_struct and autoregressvie_aug, token location information is not mandatory.
>   - For the sentence "Donald Trump owns the Trump Organization," PaDeLLM can differentiate and label the entities as "PER: Trump" and "ORG: Trump" if the de-duplication mechanism is not employed, but the token location information is losing. We recognize that including token location might improve performance and will consider this in future work, as suggested in [1].
>
> [1] A Unified Generative Framework for Various NER Subtasks (Yan et al., ACL-IJCNLP 2021)
>
> **W2-drawback of de-duplication strategy**: Yes, the de-duplication mechanism will filter this case. In our dataset, instances where two mentions appear under multiple labels are rare, as evidenced by our statistics. This is why we propose using the de-duplication mechanism. However, in real-world applications where a mention might be allowed to appear under multiple labels, we can choose not to use the de-duplication strategy. This decision is a trade-off.
>
> | Dataset | Count | Ratio|
> |---------|-------|------------|
> | ACE05   | 1     | 0.00034    |
> | ConLL03 | 1     | 0.00017    |
> | Genia   | 0     | 0          |
> | ecom    | 0     | 0          |
> | msra    | 1     | 0.00013    |
> | weibo   | 0     | 0          |
> | youku   | 2     | 0.0012     |
> | resume  | 0     | 0          |

---

### Official Review · Reviewer_GKXR · 2024-07-18

**Soundness:** 2
**Presentation:** 3
**Contribution:** 2
**Rating:** 6
**Confidence:** 4

**Summary:**

This paper proposes an interesting extension of the parallel text generation paradigm, where the authors tackle the NER task and propose to generate the labels independently. For each label prediction, the proposed method first predicts the number of mentions and then predicts the exact entity. The results show that the proposed model performs reasonably well, while achieving faster inference.

**Strengths:**

1. The proposed method is a pioneer work to accelerate LLM generation following the parallel generation paradigm.
2. We do observe significant speed-up empirically, which suggests the proposed method may be of value in real word applications.

**Weaknesses:**

1. The importance of the two-step prediction for each entity is not justified. I feel there should be a baseline such that the multiple mentions can be predicted together in an autoregressive fashion. For example, I can predict "entity type: LOC Italy English" as a whole.
2. Fundamentally, parallel predictions should be weaker than autoregressive predictions due to the drop in dependency capturing. However, we observe from Table 4 that AR models are noticeably worse than the parallel approach. Since these results contradict common wisdom, there needs more effort to justify them. For example, the authors may need to reveal the full training/testing configurations of both the AR and parallel models, and there could be some more detailed error analysis to show how AR models are making more mistakes than the parallel approach.
3. The proposed approach may face difficulty when a word is used multiple times with different types. For example, in "Washington lives in Washington," the proposed approach may predict "LOC" and "PER" for both "Washington"; however, it can not align them because the parallel approach is ordered agnostic among entities.
4. The proposed method needs finetuning to adjust the LLMs, which can be difficult when it comes to very large LLMs.

**Questions:**

1. Is the duplication issue, as mentioned in Figure 2, very common? Do you have statistics for this?
2. Do you know if the testing datasets are in the training data of the LLM?

**Limitations:**

The authors make earnest efforts to discuss the limitations. In addition, the previously mentioned Weakness 3 is another potential limitation. The authors may include further discussion in this regard.

---

> ### Author Rebuttal · Authors · 2024-08-05
>
> We thank the reviewer for the thoughtful feedback and appreciate the recognition of our paper’s contributions. We are grateful for the opportunity to address the concerns raised.
>
> **W1-Justification of the importance of two-step prediction**:
>
>  We conducted an additional experiment using one-step prediction, where all mentions of the same label are predicted in a single sequence. As shown in the results below, the inference speed of this approach falls between the two-step prediction and purely AR model, which is as expected. However, the prediction quality is lower than that of the two-step prediction. In other words, two-step prediction outperforms one-step prediction in both inference speed and prediction quality. We will include these new results in the updated version as strong justification for two-step prediction.
>
> |           Latency     | ace05 | conll03 | genia |
> |----------------|-------|---------|-------|
> | PaDeLLM_multi  | **255.53**|  **229.74**  |  **316.90**|
> | OneStep_multi  | 386.93| 272.22  | 513.63|
>
> |        f-score        | ace05 | conll03 | genia |
> |----------------|-------|---------|-------|
> | PaDeLLM_multi  | **85.02** | **92.52**   | **77.66** |
> | OneStep_multi  | 80.98 | 91.36   | 76.27 |
>
>
> **W2-Reason that AR model is worse**: We acknowledge that AR methods, which rely on dependency tokens, offer certain advantages. However, they also encounter challenges in specific scenarios:
>   - Nested NER: For example, in the ACE dataset, the term "human" is nested within the "Human rights group." All AR methods struggle to identify this nested structure. In contrast, PaDeLLM, which decomposes from dependencies, effectively addresses this issue.
>
> ||||
> |-------|---------------------------------------------------------------------------------------------------------------------------------|---------|
> | **Input** | Human rights group Amnesty International said Friday 's verdict `` represents another step in the further deterioration in the human rights situation in the country . | f-score |
> | **GT** | "PER": ["Human", "human"], "ORG": ["Human rights group", "Human rights group Amnesty International"], "GPE": ["the country"]  | 1.0     |
> | **PaDeLLM** | "GPE": ["the country"], "PER": ["human"], "ORG": ["Human rights group", "Human rights group Amnesty International"]         | 0.88    |
> | **AutoStruct** | ((ORG:Human rights group), (ORG:Human rights group Amnesty International), (GPE:the country), (VEH:null), (FAC:null), (LOC:null), (PER:null), (WEA:null)) | 0.74    |
> | **AutoAug** | [ [ Human rights group /ORG] Amnesty International /ORG] said Friday 's verdict `` represents another step in the further deterioration in the human rights situation in [ the country /GPE] . | 0.74    |
>
>   - Parse Errors in AR Output: Additionally, AR models can encounter format errors as shown below.
>     - Overlong Example (MSRA): ((组织:邓小平同志治丧委员会),(名称:江泽民),...,(名称
>       - This is due to the prediction being cut off mid-output.
>     - Error Format (Resume): ((公司:环三)+...+(名称:null)+(籍贯:null)+(学历:null)+(国籍:null))
>       - Incorrect use of "+" causes parsing issues.
>     - Invalid Null Entries: ((组织:亚协会长),...,(组织:庆祝东方新年),(组织:null),(null))
>       - Entries with "(null)" lacking an entity type lead to parsing failures.
>   - Training/test configuration: for AR models and PaDeLLM, we use the same training and testing configuration as reported in Appendix D (i.e., line 581-586)
>   - We will release all resources including inference results for the community to replicate and analyze the results. And we will add all this analysis to the updated version.
>
> **W3-same mention under multiple labels**:
>  - We acknowledge the method loses token location information, which will be explored in future work.
>  - We have identified instances where the same mention is assigned different entity types in the ground truth, as shown in the table. Although such cases are rare, addressing this issue is important for real-world applications. In these situations, we can disable the de-duplication, allowing 'Washington' to be predicted as both 'LOC' and 'PER' simultaneously.
>
> | Dataset | Count | Ratio|
> |---------|-------|------------|
> | ACE05   | 1     | 0.00034    |
> | ConLL03 | 1     | 0.00017    |
> | Genia   | 0     | 0          |
> | ecom    | 0     | 0          |
> | msra    | 1     | 0.00013    |
> | weibo   | 0     | 0          |
> | youku   | 2     | 0.0012     |
> | resume  | 0     | 0          |
>
>
> **W4-need fine-tune**:
>   - First, we tested the zero-shot generalizability of PaDeLLM, as detailed in lines 249-260, demonstrating its efficacy without further SFT to handle out-of-domain entity types, as appreciated by Reviewer XzjF.
>   - Additionally, we can employ resource-friendly fine-tuning techniques such as LoRA, P-tuning, and others to achieve the SFT.
>   - For larger LLMs, it is feasible to use training-free methods like in-context learning to enable the LLM to make the two-step prediction, as highlighted in line 287-289. This approach will be explored in future work. We will add this discussion to the updated version.
>
> **Q1**: Such situation is too rare, the statistics of prediction is shown below:
>
> | Dataset | Count | Ratio|
> |---------|-------|------------|
> | ACE05   | 22 | 0.0074   |
> | ConLL03 | 10     | 0.0017|
> | Genia   | 18     | 0.0034         |
> | ecom    | 2 | 0.0012|
> | msra    | 5     | 0.00089|
> | weibo   | 0     | 0          |
> | youku   | 3     | 0.0019     |
> | resume  | 0     | 0          |
>
> **Q2**: this is discussed in Section 6,  the primary focus of our experiments is the comparison of our proposed method with baseline methods (Auto_Struct and Auto_Aug). Given that these methods employ the same LLM as the base model, data contamination is unlikely to significantly impact the results.
>
> We kindly request your acknowledgement of our reply, and are welcome to further discussions for your questions and concerns. We would be fully appreciated if you would consider to improve the **rating**. We look forward to your response.

---

> ### Comment · Reviewer_GKXR · 2024-08-09
>
> Thanks for the responses. I have some follow-up comments/questions.
>
> **W1**
> Which decoding strategies were used in the "one-step" baselines?  How do you handle the order, i.e., "LOC Italy England" vs "LOC English Italy."
>
> **W2**
> Can you elaborate more on "All AR methods struggle to identify this nested structure? In contrast, PaDeLLM, which decomposes from dependencies?" Why is dropping dependency free from errors for the nested structure? (Additionally, can you provide a more formal definition for this nested structure?)
>
> **W3**
> Thanks for providing the statistics. However, I feel this is a fundamental limitation (in terms of capability) of the proposed method and should be addressed explicitly in the paper (at least in the Limitation section). Having post hoc refinements could be a solution, but they may also be applied to other methods, which would not be a fair comparison.
>
> **W4**
> Just to confirm: the "zero-shot generalizability of PaDeLLM" refers to the zero-shot datasets, right? The PaDeLLM model itself needs to be trained. Do you think you can make PaDeLLM a prompt-only LLM-based method?

---

> ### Author Response · Authors · 2024-08-09
> **Response for comment**
>
> Thanks for the further discussion and we are grateful for the opportunity to address the concerns.
>
> **W1-one-step pred** :
>   - In "one-step baselines". all mentions under the same label are predicted in one single sequence. If there is no label related to any mention in the example, the result is empty. Please refer to the exact prediction example below. The latency of the slowest sequence is reported as the overall latency of one example.
> | Entity | Text                                                                                                                                                                                                                                                               | NER Result |
> |--------|--------------------------------------------------------------------------------------------------------------------------------------------------------------------------------------------------------------------------------------------------------------------|------------|
> | ORG    | \<entity>ORG\<text>2004-12-20T15:37:00 Microscopic microcap Everlast, mainly a maker of boxing equipment, has soared over the last several days thanks to a licensing deal with Jacques Moret allowing Moret to buy out their women's apparel license for $$ 30 million, on top of a $$ 12.5 million payment now. | NER result: ["Microscopic microcap Everlast", "a maker of boxing equipment", "their"] |
> | PER    | \<entity>PER\<text>2004-12-20T15:37:00 ... million payment now. | NER result: ["Jacques Moret", "Moret", "their", "their women"] |
> | GPE    | \<entity>GPE\<text>2004-12-20T15:37:00 ... million payment now.  | NER result: [] |
> | LOC    | \<entity>LOC\<text>2004-12-20T15:37:00 ... million payment now.  | NER result: [] |
>
>   - The order of mentions is kept the same as in the ground truth, aligning with the data provided by the respective dataset.
>
> **W2-nested NER elaborate**:
> - Definition of nested structure:
> In the table (ground truth), the phrase 'highly diverged [Drosophila [homeodomain |DNA] |DNA]' shows a nested structure where 'homeodomain' (DNA) is nested within 'Drosophila homeodomain' (another DNA).
>
> - Elaboration
> AR methods struggle with such hierarchies due to their reliance on linear dependencies. The table's results (Auto_Aug, Auto_struct) highlight the need for models to not only **understand the input and effectively model long-range dependencies to produce the correct format to be parsed.**
> In contrast, PaDeLLM simplifies nested dependencies by breaking the tasks down during the two-step prediction, allowing it to maintain a shorter output format, making PaDeLLM better suited for handling nested structures compared to AR methods.
>
> We will add this discussion to the Appendix in the updated version.
>
> |  |  |
> | --- | --- |
> | **Input** | When the homeodomain from HB24 was compared to known mammalian and Drosophila homeodomains it was found to be only moderately conserved, but when it was compared to a highly diverged Drosophila homeodomain, H2.0, it was found to be 80% identical. |
> | **Ground truth** | When the [ homeodomain /DNA] from [ HB24 /DNA] was compared to known mammalian and Drosophila homeodomains it was found to be only moderately conserved, but when it was compared to a highly diverged [ Drosophila [ homeodomain /DNA] /DNA] , H2.0, it was found to be 80% identical. |
> | **Auto_Aug** | When the [ homeodomain /DNA] from [ HB24/DNA] was compared to known mammalian and Drosophila homeodomains it was found to be only moderately conserved, but when it was compared to a highly diverged [ Drosophila [ homeodomain /DNA] , [ H2.0, /DNA] it was found to be 80% identical." |
> | **Auto_struct** | ((DNA:homeodomain),(DNA:HB24),(DNA:homeodomains),(DNA:Drosophila homeodomain),(DNA:H2.0),(RNA:null),(cell_line:null),(protein:null),(cell_type:null)) |
> | **PaDeLLM** | str1:entity type:\nDNA\n\n<num>3\n<mention 1>homeodomain|
> ||str2:entity type:\nDNA\n\n<num>3\n<mention 2>HB24|
> ||str3:entity type:\nDNA\n\n<num>3\n<mention 3>Drosophila homeodomain|
>
>
> **W3**:We agree that this is a fundamental limitation of our proposed method. We will explicitly discuss this in the Limitation section. Additionally, we will include the experimental results without de-duplication in Table 4 (currently only presented in Table 6), to ensure a fair comparison for readers in the updated version.
>
> **W4**:Yes, the "zero-shot generalizability of PaDeLLM" refers to the zero-shot datasets, PaDeLLM model itself needs to be trained. We believe that the strong instruction-following capabilities of LLMs (especially those much larger LLMs) make it feasible to implement PaDeLLM through in-context learning or other prompt engineering techniques without additional training.
>
> If this addresses your concern, we would be fully appreciated if you would consider to improve the rating.

---

> > ### Comment · Reviewer_GKXR · 2024-08-11
> >
> > Thanks for clarifying. I'll raise my overall recommendation to 6, and I hope our discussion can help the revision.

---

> > > ### Author Response · Authors · 2024-08-11
> > > **Thanks**
> > >
> > > Thank you for the thoughtful discussion and for raising the rating. We will continue to improve the revision by taking the feedback into account.

---

### Decision · Program_Chairs · 2024-09-25

**Decision:**

Accept (poster)

**Comment:**

The paper presents a two-step parallel NER prediction scheme, which predicts the number of entities and then parallelises their label prediction.  The method achieves a significant speedup over sequential models and reasonable performance on benchmarks.  Weaknesses outlined by reviewers were all addressed during the rebuttal/discussion period.